# Risky decision-making and nonsuicidal self-injury among university students: Examining the role of criticism feedback

**Brooke H. Nancekivell**[1], **Lily W. Martin**[2], **Jill A. Jacobson**[2], **J. D. Allen**[3], **Jeremy G. Stewart**[2]*

**1** Department of Psychology, Simon Fraser University, Burnaby, British Columbia, Canada, **2** Department of Psychology, Queen's University, Kingston, Ontario, Canada, **3** Department of Psychology, University of California, Berkeley, Berkeley, California, United States of America

☯ These authors contributed equally to this work.
* jeremy.stewart@queensu.ca

**Data Availability Statement:** The de-identified data is held on Open Science Framework (OSF). The project where the data is stored is called "Risky

## Abstract

Risky decision-making putatively contributes to nonsuicidal self-injury (NSSI) yet empirical support for this association is inconsistent. Studies have not simulated socioemotional contexts most closely linked to NSSI, which may partially explain mixed findings. Accordingly, we examined the association between NSSI and risky decision-making following the receipt of criticism from a close other, a key interpersonal context. The study included 286 university students ($M_{age} = 21.11$) oversampled for a lifetime history of NSSI. Participants completed a modified Iowa Gambling Task, in which they chose to play or pass on "good" and "bad" decks associated with monetary gains and losses. Participants also completed the Criticism Gambling Task, which was identical to the Iowa Gambling Task except critical audio comments preceded each block of trials. Based on results of multilevel growth curve analyses, decreases in risky decision-making were steeper on the Iowa Gambling Task compared to the Criticism Gambling Task, suggesting poorer learning in the context of criticism. Further, how past-month NSSI was related to changes in risky decision-making across blocks differed between the two tasks, $b = -0.004$, $t(3140.00) = 2.48$, $p = .013$. On the Iowa Gambling Task, all participants decreased their risky decision-making, whereas on the Criticism Gambling Task, higher past-month NSSI frequencies were associated with riskier decision-making. Our findings support associations between risky decision-making and NSSI in negative socioemotional contexts, consistent with functional models of NSSI.

## Introduction

Nonsuicidal self-injury (NSSI) is the deliberate and direct damaging of body tissue in the absence of suicidal intent. NSSI begins in early-to-mid adolescence. Prevalence rates are highest in adolescence (~17%; [1]) and young adulthood (13–40%; [2, 3]) and then steeply reduce in later adulthood [4, 5]. NSSI predicts a range of negative psychosocial outcomes later in life [6, 7] and is among the strongest predictors of suicide attempts [8]. Thus, improved

actions, Emotions and Decision-making (RED)" and the URL for the project is: https://osf.io/fctq5/.

**Funding:** The research was supported by internal seed funding for undergraduate research projects. The funds were awarded to BHN and JGS. Funder: Arts and Science Undergraduate Society (ASUS), Queen's University Award Name: Arts and Science Undergraduate Research Fund (ASURF) Grant Number: NA (award/grant number not assigned) Award Value: $6,496 (CAD) URL: https://www.queensasus.com/asurf The funder did not play any role in the study design, data collection and analysis, decision to publish, or preparation of the manuscript.

**Competing interests:** The authors have declared that no competing interests exist.

understanding of processes that contribute to the persistence and worsening of NSSI behaviour is needed. Indeed, this knowledge may bolster early detection of NSSI and could lay a foundation for mechanism-focused interventions.

Scholars and clinicians have long theorized motivations for initiating and maintaining NSSI, a physically and psychologically costly behavior. Humans have a biological drive to survive and avoid pain, yet NSSI involves direct and self-inflicted pain [9]. It was historically assumed that people who engage in NSSI are emotionally reactive and have a fundamental inability to resist urges to self-injure (e.g., [10]). Partly due to this perspective, NSSI has been framed as a condition involving difficulties with impulse control. Impulsivity, defined as a tendency to react rashly or hastily to stimuli without considering future consequences to oneself or others [11], has accordingly received considerable empirical attention as a contributor to NSSI. Yet, cross-sectional and prospective studies of this association have frequently yielded small or nonsignificant effects [12–18]. Notably, most studies (e.g., [19]) treat impulsivity as a unitary construct, potentially occluding connections between NSSI and impulsivity's distinct facets. Extant research has also predominantly used self-report assessments that operationalize impulsivity as a trait-like dimension. Such self-report assessments are prone to response biases and may not be relevant to how impulsivity influences NSSI urges and behaviours in key contexts (see [19]). To better understand the role of impulsivity in NSSI, it is critical to focus on impulsivity-related traits most strongly tied to NSSI and related maladaptive behaviours.

Risky decision-making (i.e., disadvantageous or impaired decision-making) is a facet of impulsivity that may be particularly relevant to NSSI. Decision-making, in general, involves gathering information, assessing options, and choosing among alternatives [20]. In the context of research on NSSI, and psychopathology more broadly, risky decision-making involves mis-estimating future outcomes (e.g., "how I feel now is how I will feel in the future"), a preference for immediate rewards, and/or the continued investment in an action with a negative outcome [21, 22]. Thus, risky decision-making connotes a short-term view of situations that prioritizes temporal proximity over absolute magnitude of an action's anticipated results or "discounting" of delayed effects. This limited perspective, in turn, raises the likelihood of selecting options like engaging in NSSI that produce immediate reward and/or removal of an aversive circumstance (e.g., relief from negative emotions) despite the consequences (e.g., risk of infection), which is in line with models of NSSI that emphasize affect regulation (e.g., [23, 24].

The Iowa Gambling Task (IGT; [25]) is a commonly used behavioural measure of risky decision-making. It involves probabilistic discounting, in that some choices yield high rewards in the short term but are disadvantageous (e.g., higher net losses) over time [26]. Despite the theoretical importance of risky decision-making to NSSI behaviour (see [14, 22]), the few studies that have investigated the link between IGT performance and NSSI have yielded mixed findings. Five studies found no significant differences in IGT performance in adults ([27], Study 2; [28, 29]) or adolescents ([27], Study 1; [30]) with and without histories of NSSI. Collectively, these studies included modest samples ($N$s = 40–133; 20–64 participants reporting NSSI) and compared IGT performance between people with and without any lifetime NSSI (cf. [27], Study 2). Recently, in a larger sample of adolescents ($N$ = 240), Lutz and colleagues [31] found that, compared to those with no lifetime NSSI, youth reporting a history of repetitive NSSI (i.e., four or more episodes per year) made more risky decisions on the Cambridge Gambling Task, which involves comparable parameters to the IGT. Thus, moderating characteristics like frequency and severity may be critical for unpacking how risky decision-making contributes to NSSI.

Limited consideration of socioemotional states germane to NSSI represents another empirical gap in this literature. NSSI is used to modulate emotional experiences [9, 24], and negative affect is a reliable correlate and precipitant of NSSI [32, 33]. Interpersonally focused negative

emotional states, specifically, may be most relevant to understanding how risky decision-making contributes to NSSI [9, 34]. Indeed, interpersonal life stress (e.g., arguments) often precedes or co-occurs with NSSI [35–38]. Interpersonal emotional processes, such as rejection sensitivity [39] and anger towards others (e.g., [40]), are linked to NSSI behaviour over brief periods. NSSI may reduce negative affect, including interpersonally relevant emotions (e.g., guilt), shortly after engagement (e.g., [41–43]; c.f. [44, 45]) as well as increase positive affect [41, 43, 46]. Interpersonal contexts and the social emotions they evoke are central to NSSI. Thus, accurately simulating the socioemotional and/or contextual circumstances that tend to precede episodes of NSSI may enhance the validity of experimental investigations of risky decision-making.

Allen and colleagues [34] extended work on risky decision-making in NSSI by embedding audio criticism in the IGT that participants were to imagine receiving from a close other. Their Criticism Gambling Task (CGT) is meant to simulate socioemotional contexts with ecological relevance to NSSI. Criticism, whether perceived from others or imposed on oneself, is implicated in models of NSSI and related psychopathology [9]. Criticism from loved ones is associated with NSSI [47–49], and self-criticism may mediate this association [48, 50, 51]. As Allen and colleagues [34] contend, audio criticism on the CGT may parallel situations of heightened risk for NSSI more closely than other mood manipulations (e.g., [29]). Across two studies, Allen et al. [34] found that the CGT produced increases in negative affect. Further, risky decision-making on the CGT was associated with a greater number of NSSI episodes in the past year (Studies 1 and 2) and in participants' lifetimes (Study 2). A key limitation is that their studies did not include a control condition wherein decision-making was measured in the absence of criticism. Thus, it remains unclear whether NSSI behaviour is associated with risky decision-making exclusively in the context of criticism, or whether there is a dose-response relation between NSSI episodes and riskier decision-making in general.

## Current study

The goal of this study was to examine the role of criticism during decision-making among young adults with NSSI histories. We incorporated additional methodological and design features to extend work on this topic by addressing five main limitations. First, to address the lack of control condition in Allen and colleagues' [34] work, participants completed a modified version of the IGT [52] and the CGT [34], which allowed us to test whether (a) overall patterns of risky decision-making and (b) relations between NSSI and risky decision-making differed in the context of no criticism (IGT) versus criticism (CGT). Second, research on the link between NSSI and IGT performance has typically used a single measure of overall risky decision-making or has examined decision-making across blocks without considering the multilevel structure of the data [27, 29–31, 34]. We fit multilevel growth curve models to account for interdependence among observations and better model changes in risky decision-making over time. Third, prior work has used the classic IGT [25] wherein participants choose a card from one of four decks–two that are advantageous and two that are disadvantageous–on each trial. A drawback of this version is that participants can ignore certain decks; thus, it is unclear whether risky decisions involve selecting disadvantageous decks or not selecting advantageous decks. Accordingly, we used a play/pass version of the IGT wherein one of the four decks is highlighted on each trial, and participants make a forced decision to either play or pass on the card presented. We focused on participants' behaviour on disadvantageous decks; theoretically, playing on these decks reflects a focus on larger potential gains while ignoring or not recognizing much larger potential losses as well as average losses (i.e., negative expected values) over time [53]. Fourth, studies have not consistently tested rival predictors of IGT

performance: variables that are related to risky decision-making, such as trait impulsivity [54, 55], suicidal behaviour [22], depression symptoms, and trait self-criticism [34]. Thus, we conducted secondary analyses to examine if relations between NSSI and risky decision-making were robust when accounting for these variables. Finally, we extended work that has confined analyses to the presence or absence of lifetime NSSI (or NSSI in other time windows) by considering the recency and frequency of NSSI episodes. We examined NSSI over the past month, specifically, based on research supporting associations between past-month NSSI and a variety of psychological and psychosocial outcomes (e.g., post-hospitalization persistence of NSSI, [56]; see also [57–59]). We tested two primary hypotheses in a sample of university students who completed the IGT and CGT. First, we hypothesized that overall risky decision-making (i.e., playing on disadvantageous decks) would decrease significantly across blocks on both tasks. Further, given that laboratory stressors (e.g., audio criticism) contribute to riskier decision-making on the IGT [60] by putatively interfering with adaptive avoidance of prior negative outcomes [61], we hypothesized that decreases in risky decision-making over time would be less pronounced on the CGT compared to the IGT. Second, in line with Allen et al.'s [34] findings and prior research using the IGT (e.g., [27–30]), we hypothesized that more frequent past-month NSSI would be associated with a less pronounced decrease in risky decision-making over time, and that this effect would occur for CGT, but not IGT, performance.

## Materials and methods

### Participants

Participants were university students recruited from a course-based participant pool and online advertisements posted on social media and other forums relevant to students at our institution (e.g., closed Queen's University Facebook groups). Regardless of the recruitment avenue, eligible participants were current students at our institution with past or current NSSI and/or a past suicide attempt. For participant pool recruitment, we contacted students who reported lifetime NSSI and/or a past suicide attempt as well as students with elevated scores on measures capturing correlates of suicidal thoughts and behaviours: the Acquired Capability for Suicide Scale-Fearlessness About Death [62]; the Reasons for Living Inventory-Fear of Suicide scale [63]; and the Painful and Provocative Events Scale-Revised, [64]. See https://osf.io/fctq5/ for more details on recruitment procedures. For online advertisements, students were encouraged to contact our lab if they met eligibility criteria and were interested in participating.

Of the 372 university students who completed the study, 66 were excluded because of incomplete IGT and/or CGT data. Specifically, 17 participants did not access the portion of the study where the IGT/CGT were completed (i.e., no task data), and 49 participants completed some but not all the tasks (most of these participants had 50% or less task data). An additional 20 participants were excluded because their NSSI and/or suicide attempt history was unknown. We removed participants with missing IGT/CGT and/or self-injury history data because these variables were needed for primary analyses and in most cases, the amount of missing data was substantial. Thus, the final sample was 286 participants; the demographic and self-injury history characteristics of the sample are summarized in Table 1.

### Measures

**Iowa Gambling Task (IGT) and Criticism Gambling Task (CGT).** We used the modified IGT [52] and CGT [34] to assess risky decision-making. Participants began each task with $2000 in virtual money and were instructed to win as much money as possible. Each task had six blocks with 20 trials per block (120 trials total). In each trial, participants were presented with four decks of cards, one of which had an arrow above and was the *target* card. Participants

**Table 1. Demographic and self-injury history characteristics for the sample.**

|  | *M* or *n* | *SD* or % |
|---|---|---|
| **Age** | 21.11 | 6.20 |
| **Gender** |  |  |
| Woman | 238 | 83.22 |
| Man | 45 | 15.73 |
| Nonbinary | 1 | 0.35 |
| **Race/Ethnicity** |  |  |
| Asian | 51 | 17.83 |
| Black or African | 3 | 1.05 |
| Hispanic, Latino, or Spanish | 1 | 0.35 |
| Middle Eastern or North African | 7 | 2.45 |
| White | 179 | 62.59 |
| More than one race/ethnicity | 36 | 12.59 |
| Race/ethnicity not listed | 4 | 1.40 |
| **Self-injurious behaviours** |  |  |
| Past-week NSSI | 29 | 10.14 |
| Past-month NSSI | 49 | 17.13 |
| Past-year NSSI | 99 | 34.62 |
| Lifetime NSSI | 159 | 55.59 |
| Age of NSSI onset (years) | 14.15 | 3.13 |
| Duration of NSSI engagement (years) | 4.64 | 4.50 |
| Number of NSSI methods | 1.45 | 1.65 |
| Lifetime suicide attempt | 67 | 23.43 |

*Note.* NSSI = nonsuicidal self-injury. *N* = 286. Three participants preferred not to indicate their race/ethnicity and two participants were missing gender and race/ethnicity data. Note: For self-injurious behaviours, all timeframe variables (past-week, -month, -year, and lifetime) are *n*s (i.e., refer to the number of people who reported NSSI and/or suicide attempt over the specified timeframe).

had four seconds to decide to *play* or *pass* on the target card. If they played, they either won money, lost money, or neither won nor lost money. If they passed, they neither won nor lost money. The four decks had various winning probabilities: two *disadvantageous* decks yielded higher wins but even higher losses, and two *advantageous* decks yielded smaller wins but even smaller losses. Participants were told that some decks were more profitable than others. Participants with no NSSI history generally improve over the course of the IGT (i.e., learn the properties of the decks and make fewer selections from disadvantageous decks; [52, 65]).

The CGT was identical to the IGT except before the task, participants selected a person with whom they had a close relationship (i.e., mother, father, other relative). At the start of each of the six blocks, participants listened to a 20-second audio clip of critical comments spoken by a female voice [34] and were to imagine that the person they selected was saying the comments to them. Most participants chose their biological mother (*n* = 262, 91.61%). The six clips were originally based on recordings from a study where mothers were asked to record critical comments about their daughters who later listened to the recordings during a functional neuroimaging procedure [59, 60, 66, 67]. An example of a critical comment that participants heard was: "One thing that really bothers me about you is that you always have to get your own way. You have a hard time taking 'no' for an answer, and you really can get resentful when you don't get what you want. You don't seem to realize that there needs to be some give and take if you're going to get along with people. You have a lot of trouble with your

relationships, and this is one of the reasons why." The audio comments we used were identical to Allen and colleagues' [34] stimuli, and the authors reported that the audio criticism produced significant increases in negative affect in Study 1 ($\eta_p^2$ = 0.40) and Study 2 ($d$ = 0.24). Importantly, the effect of criticism on increased negative affect was not qualified by interactions with NSSI features [34]. Generally, audio criticism reliably increases negative mood and reduces positive mood, producing activation in brain regions involved in emotional processing [66–69].

In both tasks, we focused on risky decisions, defined as plays on disadvantageous decks, which were possible on trials in which a disadvantageous deck was presented (10 of 20 trials per block). The outcome was the proportion of risky decisions on each block, ranging from 0 (0 of 10 disadvantageous deck trials) to 1 (10 of 10 disadvantageous deck trials).

## Self-Injurious Thoughts and Behaviours Interview-Revised (SITBI-R)

We assessed self-injurious behaviours with the online self-report version of the Self-Injurious Thoughts and Behaviours Interview-Revised (SITBI-R; [70]). See Table 1 for descriptive statistics on participants endorsing self-injurious behaviours (NSSI and suicide attempts). See Table 2 for descriptive statistics on NSSI frequencies. Of participants who had engaged in NSSI, the most common method was cutting or carving skin ($n$ = 118, 74.21%), followed by self-hitting ($n$ = 95, 59.75%), and scraping skin ($n$ = 85, 53.46%). Thirty-five participants reported receiving treatment for their NSSI, 25 (15.72%) psychological intervention, and 10 (6.29%) medical intervention.

**Table 2. Bivariate correlations and descriptive statistics for study variables.**

| | 1 | 2 | 3 | 4 | 5 | 6 | 7 | 8 | 9 | 10 |
|---|---|---|---|---|---|---|---|---|---|---|
| **Bivariate Correlations** | | | | | | | | | | |
| 1. IGT average | | | | | | | | | | |
| 2. CGT average | 0.57*** | | | | | | | | | |
| 3. Past-week NSSI | -0.04 | -0.01 | | | | | | | | |
| 4. Past-month NSSI | -0.01 | 0.05 | 0.78*** | | | | | | | |
| 5. Past-year NSSI[1] | 0.00 | 0.12* | 0.55*** | 0.72*** | | | | | | |
| 6. Lifetime NSSI[2] | -0.07 | 0.02 | 0.40*** | 0.51*** | 0.73*** | | | | | |
| 7. Lifetime SA | -0.06 | 0.05 | 0.12 | 0.12* | 0.22*** | 0.41*** | | | | |
| 8. Recent depression symptoms | -0.07 | 0.04 | 0.31*** | 0.31*** | 0.35*** | 0.31*** | 0.09 | | | |
| 9. Trait self-criticism | 0.00 | 0.01 | 0.27*** | 0.30*** | 0.36*** | 0.35*** | 0.16 | 0.69*** | | |
| 10. Trait impulsivity | 0.17** | 0.14* | 0.24*** | 0.25*** | 0.29*** | 0.21*** | 0.13* | 0.34*** | 0.30*** | |
| **Descriptive Statistics** | | | | | | | | | | |
| Mean / $N$ | 0.66 | 0.67 | 0.16 | 0.69 | 6.72 | 83.34 | 67[3] | 15.50 | 31.81 | 43.47 |
| $SD$ | 0.19 | 0.19 | 0.63 | 2.06 | 20.41 | 309.94 | – | 6.02 | 10.77 | 8.59 |
| Range | 0–1 | 0–1 | 0–7 | 0–15 | 0–156 | 1–3000 | – | 6–30 | 8–57 | 25–71 |
| Skewness | -0.54 | -0.73 | 6.31 | 3.86 | 4.77 | 6.95 | – | 0.27 | -0.11 | 0.29 |
| Kurtosis | 0.35 | 1.37 | 53.53 | 16.63 | 25.48 | 57.00 | – | -0.65 | -0.59 | -0.05 |

*Note*. IGT = Iowa Gambling Task; CGT = Criticism Gambling Task; NSSI = nonsuicidal self-injury; SA = suicide attempt; $SD$ = standard deviation.

\*$p < .05$

\*\*$p < .01$

\*\*\*$p < .001$. $N$ = 286 unless otherwise indicated.

[1]$N$ = 284

[2]$N$ = 277

[3]Refers to the number of participants who reported at least one lifetime suicide attempt.

Only past-month NSSI frequency was used as a predictor in multilevel growth curve models. The rationale was two-fold. First, we wanted to prioritize recent NSSI because our outcome variable was state-like (i.e., performance on a task). Very few participants endorsed past-week NSSI (only 10 had >1 past week episode); thus, past-month NSSI balanced recency and practicality (i.e., a predictor with significant variability). Second, past-month NSSI is prospectively linked to important outcomes like rehospitalization and NSSI persistence [56–59].

**Patient-Reported Outcomes Measurement Information System (PROMIS).** We assessed recent depression symptoms with items from the Patient-Reported Outcomes Measurement Information System (PROMIS; [71]) depression item bank (Depression Short Form 6a; [72]. Participants rated the severity of six depression symptoms over the past seven days from 1 (*never*) to 5 (*always*). We summed scores across items, where higher scores indicated a greater severity of depression symptoms over the past week. Consistent with prior work [72], internal consistency reliability of the PROMIS Depression Short Form 6a was excellent in our sample, ω = .96. See Table 2 for descriptive statistics on PROMIS scores.

**Self-Rating Scale (SRS).** We assessed trait self-criticism with the Self-Rating Scale (SRS; [73]). Participants rated the extent to which they agreed with eight items assessing the emotional and cognitive aspects of self-criticism from 1 (*strongly disagree*) to 7 (*strongly agree*). We summed scores across items, where higher scores indicated a greater degree of trait self-critical beliefs. Consistent with prior work [34, 73–76], internal consistency reliability of the SRS was good in our sample, ω = .90. See Table 2 for descriptive statistics on SRS scores.

**Short UPPS-P impulsive behaviour scale.** We assessed trait impulsivity with the Short UPPS-P Impulsive Behaviour Scale (S-UPPS-P; [77]), a shortened version of the UPPS-P Impulsive Behaviour Scale [19]. Participants rated the extent to which they agreed with 20 items assessing five facets of trait impulsivity (four items each): sensation seeking, positive urgency, negative urgency, lack of premeditation, and lack of perseverance from 1 (*agree strongly*) to 4 (*disagree strongly*). The sensation seeking, positive urgency, and negative urgency subscales were reverse coded. We summed scores across subscales. Internal consistency reliability for the overall S-UPPS-P was adequate in our sample, ω = .84. See Table 2 for descriptive statistics on Short UPPS-P scores.

## Procedure

Interested participants who provided contact information relevant to safety procedures (email, phone number, country completing the study in) were sent the Qualtrics[TM] [78] study link that included the Letter of Information and Consent Form. Participants who provided electronic written consent completed a battery of questionnaires in Qualtrics[TM] [78] and the IGT and CGT in Millisecond's Inquisit [79]. Participants were randomly assigned to one of two administration orders. In order one, participants completed the questionnaires first, followed by the IGT and CGT; order two was the reverse. Within each administration order, the sequence of questionnaires was randomized, as was the order of the IGT and CGT. Participants then completed a positive mood induction, which included listening to Mozart's "A Little Night Music" and reading positive Velten statements [80]. Finally, participants read the Debriefing Form, which included a list of local, national, and international mental health resources. Participants were compensated with either academic credits or a gift card. The study was approved by the Queen's University General Research Ethics Board (GREB).

## Results

All statistical analyses were conducted in R version 4.3.0 [81]. We used the following packages: tidyverse [82] and reshape2 [83] for data wrangling; lme4 [84], lmerTest [85] performance

[86], and interactions [87] for the multilevel growth curve analyses; and psych [88] and ggplot2 [89] for data visualization.

## Correlation analyses

We first performed correlation analyses to examine the zero-order relationships among key variables. As expected, average IGT and CGT performances (i.e., average proportion of risky decisions across the six blocks) were positively and substantially associated with each other, and each was positively associated with trait impulsivity albeit weakly. The only significant relationship between task performance and any NSSI or past suicide attempt was a small positive association between average CGT performance and past-year NSSI. All timeframes of NSSI frequency (i.e., number of episodes over the past week, month, year, and lifetime) were moderately to highly correlated with each other but were less strongly related to having a past suicide attempt. All the NSSI variables were positively related to recent depression symptoms, trait self-criticism, and trait impulsivity. See Table 2 for complete results.

## Growth curve analyses

We performed multilevel growth curve analyses to investigate trajectories of risky decision-making on the IGT and CGT. We first ran a null model with participant as the Level 2 cluster variable. The intercept for the null model was significant, $b = 0.669$, $t(285.00) = 66.74$, $p < .001$. The intraclass correlation was .428, suggesting that within a given participant the correlation between performance on any two blocks was .428, highlighting the importance of accounting for within-participant interdependence.

We then tested the two-way interaction between block and task to examine overall risky decision-making. Block was treated as a continuous variable from 0 (block 1) to 5 (block 6). Task was contrast coded to compare the IGT (coded as 0.5) to the CGT (coded as -0.5). The block x task interaction was significant, $b = -0.013$, $t(3143.00) = -3.55$, $p < .001$, suggesting that how risky decision-making changed across blocks differed on the IGT and CGT. We followed up the block x task interaction with simple slopes analysis (see Fig 1). The simple block slopes were significant and negative for both the IGT, $b = -0.021$, $t(3143.00) = 8.044$, $p < .001$, and CGT, $b = -0.008$, $t(3143.00) = -3.018$, $p = .003$. However, the block effect was significantly stronger for the IGT than CGT, such that the decrease in risky decision-making was significantly steeper on the IGT than CGT.

Next, we tested the three-way interaction between block, task, and past-month NSSI on risky decision-making. Block and task were treated the same as in the two-way interaction; past-month NSSI was treated as a continuous (uncentered) predictor. The block x task x past-month NSSI interaction was significant, $b = -0.004$, $t(3140.00) = -2.48$, $p = .013$, suggesting that how past-month NSSI was related to changes in risky decision-making across blocks differed on the IGT and CGT. We followed up the block x task x past-month NSSI interaction using the Johnson-Neyman technique to determine the regions of significance (see Fig 2). For the IGT, the simple block slope was significant and negative at all NSSI frequencies (100% of the sample). For the CGT, the simple block slope was significant and negative at NSSI frequencies $\leq 1$ (88.46% of the sample). Participants with NSSI frequencies $\geq 2$ (11.54% of the sample) did not significantly increase or decrease their risky decision-making across blocks of the CGT.

To examine the robustness of the effect of past-month NSSI, we tested rival predictors that could explain the association between past-month NSSI and risky decision-making. Specifically, we tested lifetime suicide attempts, recent depression symptoms, trait self-criticism, and trait impulsivity as interaction covariates [90, 91] in separate models. Each model included the

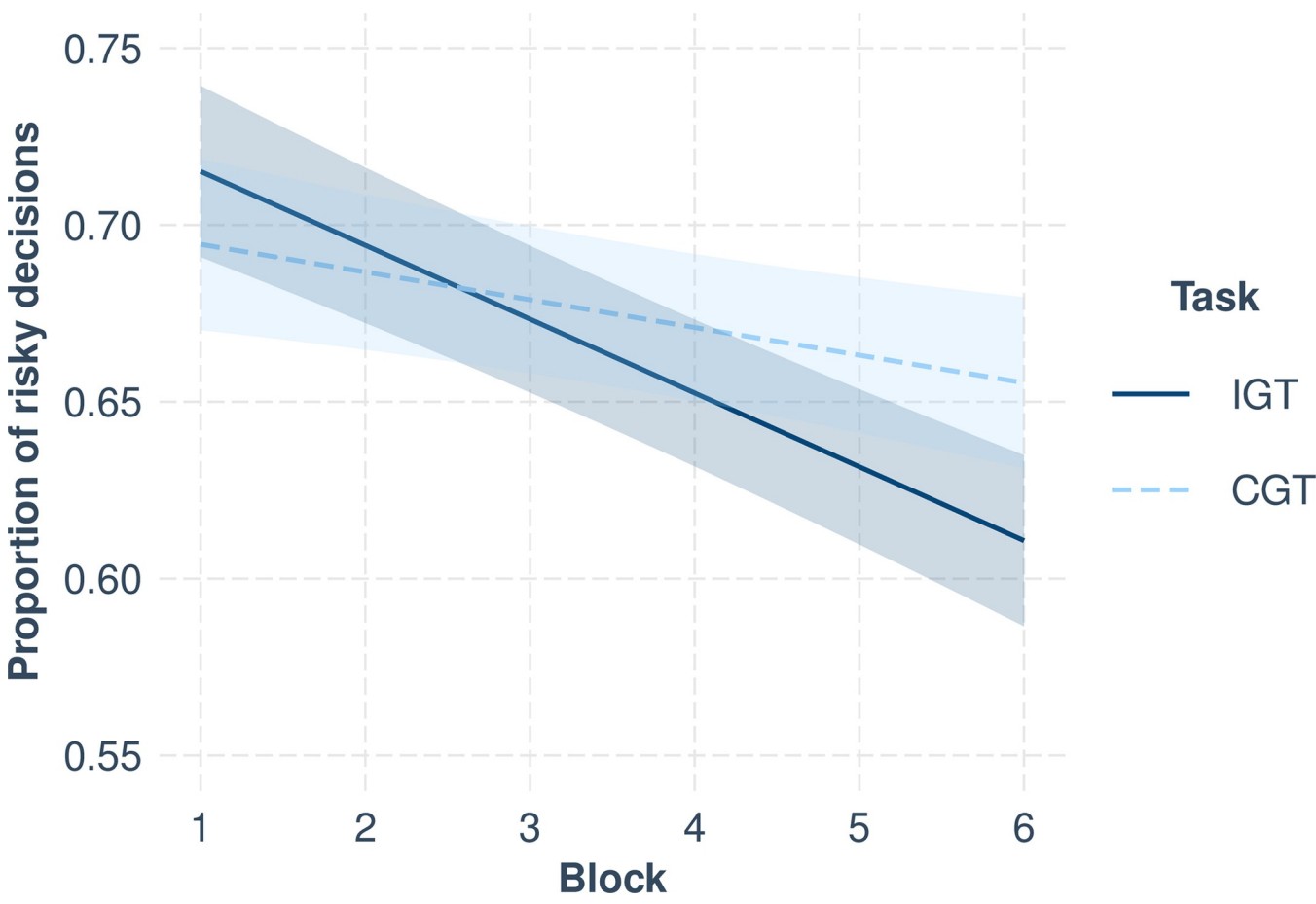

**Fig 1. Simple block slopes on proportion of risky decisions based on task.**

original predictors plus the three-way interaction between block, task, and the alternative predictor and its lower order terms. The three-way interaction involving past-month NSSI persisted beyond the effect of the potential confounding constructs: lifetime suicide attempts, recent depression symptoms (grand mean centered), trait self-criticism (grand mean centered), and trait impulsivity (grand mean centered), $p$s ≤ .016. The three-way interactions involving lifetime suicide attempts, recent depression symptoms, and trait impulsivity were not significantly associated with risky decision-making, $p$s ≥ .442. The three-way interaction involving trait self-criticism was significantly associated with risky decision-making, $b$ = 0.0008, $t$(3137.00) = 2.32, $p$ = .020. However, this interaction appeared to be the product of a suppression effect in our model (see [92]); the same three-way interaction involving trait self-criticism was nonsignificant ($p$ = .097) in a model that did not include the block x task x past-month NSSI interaction. Since the effect of past-month NSSI persisted beyond the effect of trait self-criticism, we do not present the follow-up Johnson-Neyman results for trait self-criticism. See S1 Appendix for these results.

## Discussion

NSSI is prevalent among adolescents and young adults and is concurrently and prospectively associated with negative mental health and psychosocial outcomes. Prior experimental work has proposed that risky decision-making–a tendency to make choices that may have

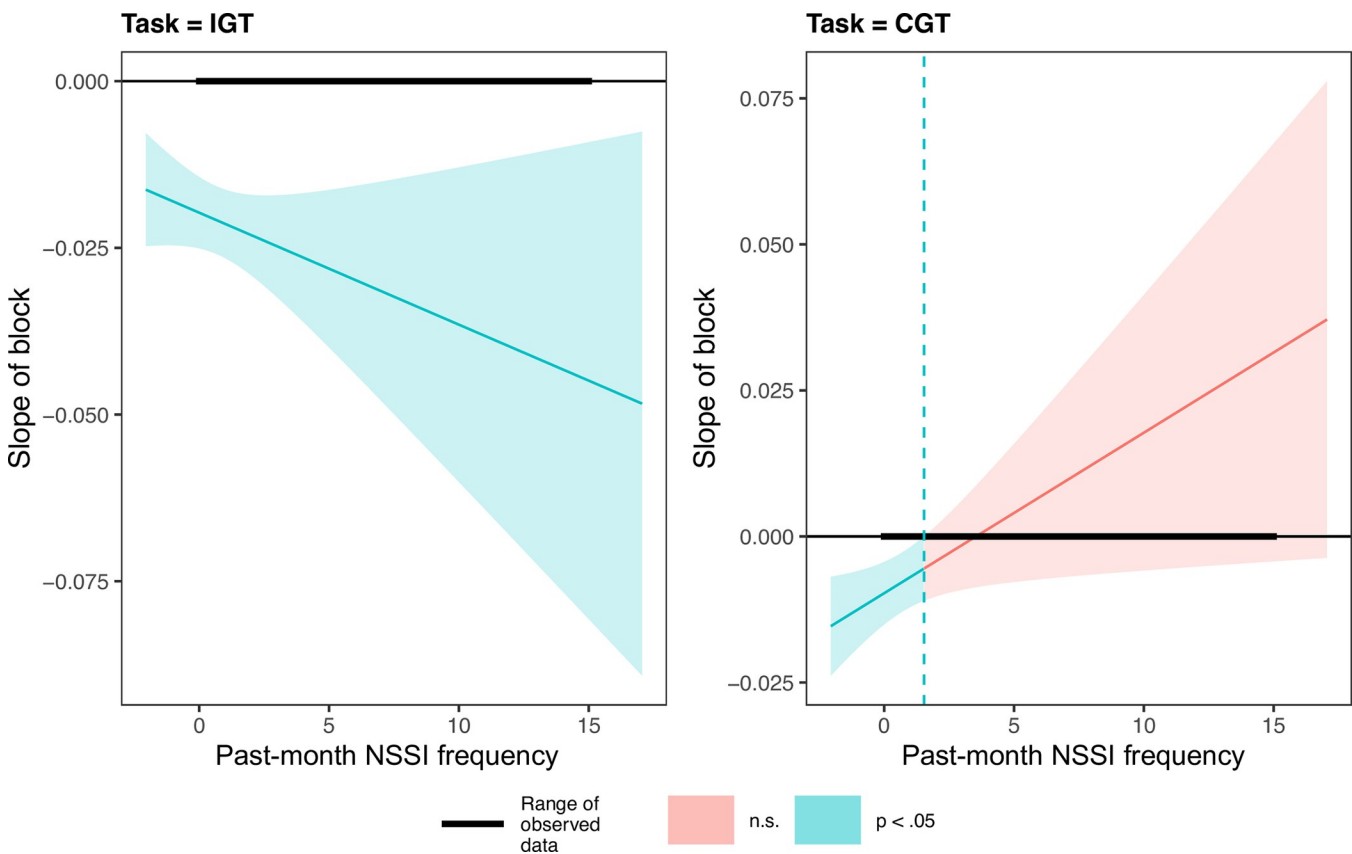

**Fig 2. Regions of significance for the simple block slopes on proportion of risky decisions as a function of past-month nonsuicidal self-injury (NSSI) and task.**

immediate benefits but can incur large costs and result in poor long-term outcomes–may contribute to NSSI behaviour. To better understand the association between risky decision-making and NSSI, we simulated and examined the effect of criticism, a socioemotional context theoretically and empirically germane to NSSI. Specifically, we administered the IGT and CGT to a sample of young adults, more than half of whom reported lifetime NSSI, and incorporated design features that addressed methodological limitations of past research.

We had three main findings. First, consistent with hypotheses and past research, risky decision-making (i.e., the proportion of plays on disadvantageous decks) decreased across blocks on both the IGT and CGT. However, this decrease over time was significantly steeper on the IGT compared to the CGT. Second, as expected, NSSI frequency in the past month moderated changes in risky decision-making across blocks on the CGT but not on the IGT. On the CGT, participants with one or no prior NSSI episodes in the past month significantly reduced their risky decision-making across the blocks, whereas participants with multiple NSSI episodes in the past month did not significantly change their patterns of risky decision-making over time. Third, our NSSI results cannot be explained by rival clinical predictors associated with risky decision-making.

## Criticism and decision-making

We observed overall patterns of decision-making on the IGT and CGT consistent with prior research. On the classic IGT, normative performance involves a linear increase in net score

(i.e., increasing the ratio of advantageous to disadvantageous deck selections) across blocks (e.g., [93]). Following the ambiguity of the early blocks, participants learn the relative risks and rewards for each deck, which drives improved performance [65]. Prior research using the play/pass IGT [52] has similarly shown that disadvantageous decision-making (selecting "bad" decks) linearly decreases across blocks in young adult samples. The proportion of plays on disadvantageous decks in our sample approximated rates in these prior studies [53, 94]. Thus, our remote administration of the play/pass IGT yielded general patterns that were like findings from in-person studies.

Participants selected disadvantageous decks less often over time in both tasks, but the decrease was significantly less pronounced on the CGT. Prior work has demonstrated that acute laboratory stressors (e.g., cold pressor) contribute to small but significant ($d = 0.26$) increases in risky decision-making on the classic IGT (see [60]). Indeed, stress may interfere specifically with adaptive avoidance of negative outcomes [61]. However, research employing socioemotional stressors has been mixed. Studies using social exclusion [29] and negative mood inductions [95] have found no evidence of differential IGT performance across conditions hypothesized to impact decision-making. The effect of acute socio-evaluative stress (e.g., the Trier Social Stress Test; [96]) on overall IGT performance has also ranged from non-significant to producing modest lower net decision-making scores (see [60]).

A reason why the CGT's audio criticism may be particularly disruptive to decision-making relative to other stress manipulations is it evokes an imaginal situation where a close relationship is threatened. Adolescence and early adulthood are characterized by heightened neurobiological sensitivity to interpersonal feedback from close others (see [97]). Further, compared to non-interpersonal stressors of similar severity, events involving rejection, criticism and/or relationship losses are more strongly linked to psychopathology and maladjustment (e.g., [98–100]). Thus, relative to other manipulations, hearing incisive criticism and imagining this coming from a close tie may be particularly salient (e.g., [47]), and thus more disruptive to avoiding choices linked to poor outcomes.

In our design, we exchanged standardization (i.e., all participants hear the same criticism) for potentially lower ecological validity. We did not measure how much participants identified with the criticism, nor did we assess participant's success with imagining the criticism coming from a loved one. Thus, although in line with prior research [34, 48, 49] we speculate that criticism is particularly relevant, there are alternative explanations for the effects we observed. We used criticism stimuli that were identical to prior CGT research and that have shown to evoke increases in negative affect [34]. The audio criticism may simply evoke more negative affect because the content and tone of the message is negative. Criticism may have functioned like negative mood inductions used in prior studies [95], and negative mood states may be more likely to disrupt decision-making among people with versus without recent NSSI [9, 32, 33]. Future research should investigate alternative explanations for our effects by, for instance, assessing areas in which participants are most often criticized *in situ*, and tailoring audio criticism stimuli to enhance participant identification with the criticism.

Although we did not examine potential mechanisms for why criticism may affect risky decision-making, Allen and colleagues [34] proposed that being criticized may activate self-critical beliefs (see [101, 102]) that impact performance. Prior research suggests that self-criticism may affect learning (e.g., determining which decks to play on more versus less) through decreased goal motivation. Indeed, people higher in trait self-criticism are less motivated to pursue goals and make less progress on their "real-life" goals over time [103–105]. Further, one experimental study found that when negative feedback threatened self-worth, it impacted participants' ability to process information that was essential to learning and succeeding on a task [106]. We did not find significant bivariate associations between trait self-criticism (SRS scores) and

average risky decision-making. Further, the interaction effects involving SRS scores were non-significant in models that did not also include NSSI's interactions with task and block. None-theless, future research that measures changes in processes like motivation and self-criticism repeatedly across the CGT is needed to delineate the mechanisms through which criticism may influence decision-making.

## NSSI and risky decision-making

Prior studies using the classic IGT have generally not found an association between NSSI and average risky decision-making [27–30]. We replicated and extended these findings using a play/pass IGT [52] that provided the advantage of attenuating the impacts of indi-vidual differences in search strategies across decks in earlier trials [53]. The play/pass IGT also allowed us to focus on disadvantageous decks, specifically, which may better capture aspects of risky decision-making (e.g., preference for immediate rewards; persisting with an action linked to negative outcomes) associated with psychopathology. Indeed, behaviour on *advantageous* decks may not change across blocks [94]; focusing on disadvantageous decks may yield a stronger signal to quantify risky decision-making. With these methodological advantages, our findings add to mounting evidence that NSSI is not associated with more risky decision-making overall across situations. Instead, a maladaptive preference for sub-optimal choices may apply *specifically* to contexts wherein people select NSSI to address a problem or challenge (see [29]).

Some of our findings were incongruent with prior work using the CGT [34]. We did not find consistent correlations between NSSI frequency and average CGT performance. The rea-son for this discrepancy may have been methodological. Our CGT added criticism to the play/pass IGT, whereas Allen and colleagues [34] created their CGT based on the classic IGT that examines net scores. Scholars have encouraged moving beyond the net score for the IGT because of the loss of information about temporal patterns across the task (e.g., [107]). Further, Allen and colleagues [34] recruited participants endorsing a lifetime history of NSSI, with pref-erence for people with substantial NSSI histories (e.g., >10 episodes), whereas our sample included some people with no lifetime NSSI. We replicated Allen and colleagues' [34] signifi-cant bivariate correlation between average CGT performance and past-year NSSI but found no significant associations between average CGT performance and past-week, past-year, or lifetime NSSI. Allen and colleagues' [34] samples had higher rates of NSSI endorsement for all time periods, and their sample's mean NSSI frequency was also higher (e.g., 40.89 episodes ver-sus 6.72 episodes for past year). Future research is needed to extend our methodology and ana-lytical approach to samples wherein NSSI is more common and severe. This approach would shed light on the discrepancies observed between our findings and prior work.

We found that people with recent, recurrent (2 or more episodes) NSSI, but not people with less frequent NSSI, did not significantly adjust their behaviour on disadvantageous decks on the CGT. Our findings align with theory and empirical evidence that NSSI is used to allevi-ate negative affect [9, 41–43]. In this context, NSSI is an appealing decision in the short-term (e.g., provides relief) but can come with large long-term costs. Criticism from close others may evoke especially salient negative, interpersonally relevant emotions [34, 37, 38]. Consequently, this socioemotional context may raise the likelihood of risky decision-making (i.e., choosing NSSI) among people with NSSI histories. As this study is cross-sectional, we cannot determine if risky decision-making is a cause or consequence (or both) of NSSI. Longitudinal research has found bidirectional associations between NSSI and lower emotion regulation over multiple time points [108]. Thus, to deepen our understanding of how risky decision-making may con-tribute to NSSI, relations between these constructs should be tested in prospective designs.

## Rival clinical predictors

We examined clinical variables that have been theoretically and empirically implicated in risky decision-making. Unexpectedly, lifetime suicide attempts, recent depression symptoms, and trait self-criticism were not associated with average IGT or CGT performance. Major Depressive Disorder is associated with lower net scores (more risky decision-making on average) on the classic IGT [109], but results for the effect of depression symptoms are more mixed (e.g., [34, 110–112]). Notably, a study that used the play/pass version of the IGT also found that recent depression symptom severity was not associated with choices on disadvantageous decks [86].

Similarly, studies reporting effects of lifetime suicide attempts on IGT performance have most often used net scores [22, 113]. Suicidal behaviour has been linked to anhedonia and blunted reward responsiveness in prior research. Perhaps poorer overall performance (net score) on the IGT is driven primarily by learning to choose advantageous decks, rather than learning to avoid disadvantageous decks. Therefore, our IGT–which focused on responses to disadvantageous decks–may not have been sensitive to differences between people with versus without prior attempts. The play/pass version of the IGT/CGT provides a tool for future research to unpack the differential contributions of approach and avoidance behaviours to decision-making (see [94]).

People who engage in NSSI may have higher self-criticism than people who do not engage in NSSI [48, 73, 112], and self-criticism is associated with more frequent and severe NSSI [113]. In line with these findings, we found that trait self-criticism was positively associated with past-week, -month, -year, and lifetime NSSI. A limitation of our approach is that we did not assess how much participants identified with the audio criticism. However, trait self-criticism is hypothesized to develop from negative experiences with key attachment figures, especially parents [114, 115], and it is consistently correlated with perceptions of how critical others are (e.g., [116]). Consistent with Allen and colleagues [34], trait self-criticism was not bivariately associated with average risky decision-making. Further, the effect of NSSI persisted when we controlled for the three-way interaction involving trait self-criticism. The latter was also nonsignificant in models that did not include NSSI. Taken together, our findings indirectly suggest that individual differences in participants' identification with the audio criticism had a modest influence on the NSSI findings, at most. Nonetheless, future experimental research designed to temporarily manipulate self-worth (e.g., [117]) and capture individual differences in participants' appraisals of audio criticism, could help clarify links among NSSI, trait self-criticism, and risky decision-making.

Trait impulsivity was significantly associated with greater average risky decision-making on both the IGT and CGT. Studies examining trait impulsivity and IGT performance using the S-UPPS-P or UPPS-P have sometimes (e.g., [54, 55]) but not always (e.g., [118, 119]) found significant associations. Scholars have frequently observed that self-report and behavioural measures of impulsivity share little variance, and measure distinct constructs (e.g., [120]). Although statistically significant, correlations in our sample among trait impulsivity and average task performance were small and may have occurred because we oversampled for self-injurious behaviours (e.g., [34]). Our primary analyses were robust to including rival predictors that were associated with NSSI, which suggests that the NSSI effects were unlikely due to confounding of clinical correlates shared among NSSI and risky decision-making.

## Limitations

Our findings need to be considered in the context of several limitations. First, as mentioned above, risky decision-making and key predictor variables were measured concurrently. Data

on how individual differences in key cognitive systems predict self-injurious behaviours over time is rare [121]. Longitudinal studies would provide a stronger test of hypothesized socioe-motional contexts that might drive a relation between risky decision-making and NSSI (e.g., criticism and negative affect; [9, 34]). Second, we used a sample of university students, many of whom were participating for course credit. NSSI is common among post-secondary students [122] and is a major source of disability and impairment in this population [123]. None-theless, our sample reflected the demographic composition of the institution as a whole: participants were predominantly White women in their early 20s. Future research should test the generalizability of our effects to samples more representative of young adults from the general population. Third, studies have shown that online administration of the IGT has good convergent validity with tasks designed to assess processes underlying IGT performance administered in-lab [124]. Perhaps because the study was conducted online, however, many participants were missing IGT/CGT and/or questionnaire data, which reduced our sample size. Research has demonstrated that the validity of data obtained in cognitive/perceptual experiments does not differ significantly from data obtained in traditional lab settings [125]. Nevertheless, future studies may benefit from in-lab administration of the tasks to improve participant retention and test for differences between remote and in-person formats.

Fourth, in the CGT, participants heard 20-second audio clips at the beginning of each block (6 total), but the IGT did not have any audio preceding the blocks. Accordingly, the tasks differed in the presence versus absence of audio messages in general. Our effects may have occurred because criticism, specifically, impacts decision-making among people with recent NSSI, but our design cannot rule out some alternative explanations. For instance, relative to the CGT, the IGT had a lower minimum time between blocks. Greater working memory capacity is associated with better performance on the IGT because working memory predicts being certain of which decks are "good" and which are "bad" (i.e., performance in the conceptual period) ([126, 127]; although see [128] for contradictory findings). Having a 20-second pause may have introduced a greater working memory load on the CGT relative to the IGT. Relatedly, the CGT may have been more distracting than the IGT; thus, greater proneness to distraction among people with recent NSSI (e.g., [129]) could have partially explained the link between NSSI and riskier decision-making on the CGT. However, trait impulsivity is associated with both lower performance on working memory tasks (e.g., [130]) and proneness to distraction (e.g., [131, 132]). We found that the interactions involving past-month NSSI in our models persisted when the effects of trait impulsivity were included. Nonetheless, future studies should use versions of the IGT that have non-critical audio feedback (e.g., neutral or positive) matched for duration to the CGT to support stronger inferences about the unique role criticism might play in how NSSI impacts risky decision-making.

Fifth, as we describe above, we did not measure the extent to which participants identified with the audio criticism they were hearing, nor did we evaluate their ability to imagine their identified close other giving this criticism. Thus, for some participants, audio criticism may have seemed less threatening or impactful; were audio criticism more tailored to what participants experience *in situ*, we may have observed larger effects. Prior research has examined criticism by having participants' parent record critical comments specifically about them [66, 68]. Adopting a similar design for future research on the CGT would address open questions about variability in the meaning participants ascribed to the audio criticism in our study.

Sixth, we used a modified version of the IGT (play/pass) and focused on disadvantageous decisions. There are several variations of the IGT that are sensitive to the differential impact of reward and punishment sensitivity on outcomes through their payoff structure (i.e., ABCD and EFGH; [133]). Previous research has shown participants with depression [134] and people with recent suicide attempts [135] may perform differently on the ABCD (reward sensitivity)

and EFGH (punishment sensitivity) versions of the IGT. Our results may not generalize to other versions of the IGT, and replications of the effect of audio criticism in ABCD and EFGH versions of the CGT, for example, would advance knowledge regarding potential moderators risk-taking in laboratory tasks. Finally, NSSI-related effects on risky decision-making (measured by the CGT) may be specific to people with recent NSSI. In our sample, recent NSSI behaviour was rare (~11%), and our effects warrant replication in samples with more severe clinical profiles (e.g., recently hospitalized patients; [136]) wherein current NSSI would be more likely.

## Conclusion

Scholars have viewed NSSI as a behaviour stemming primarily from difficulties with impulse control. Accordingly, experimental research has focused on understanding how risky decision-making may contribute to NSSI episodes. We extended prior research by examining how NSSI is related to risky decision-making in the presence and absence of audio criticism. In doing so, we mirrored a socioemotional context that has been consistently implicated in NSSI episodes in past research. We found that recent NSSI was associated with risky decision-making only in the presence of criticism, and rival clinical correlates of NSSI were not related to this pattern of decision-making performance. Our findings add nuance to theoretical models of NSSI that highlight the central roles of impulsivity (e.g., [16]). Specifically, they suggest that NSSI may be linked to certain behavioural expressions of impulsivity (i.e., risky decision-making) in relevant socioemotional contexts (being criticized [e.g., [9]]). Methodologically, they underscore the importance of incorporating ecologically relevant contexts into experimental research on putative NSSI mechanisms. In the long-term, our results may accelerate the identification of behavioural markers related to the severity and persistence of NSSI in young people.

## Supporting information

**S1 Appendix. Follow-up analyses for the three-way interaction involving trait self-criticism, block, and task in the multilevel growth curve analyses examining rival predictors.** (DOCX)

## Acknowledgments

The authors acknowledge the following people for their assistance with recruitment and data collection for this project: Jasmine Chananna, Gabrielle Craddock, Ashley Filion, Zainab Hassan, Isabelle Hau, Anjalika Khanna Roy, Jessica Mahadeo, Geneva Mason, Ibukunoluwa Okusanya, Megan Rowe, and Grace Rowed.

We thank our participants for their engagement with this research and their thoughtful feedback.

## Author Contributions

**Conceptualization:** Brooke H. Nancekivell, J. D. Allen, Jeremy G. Stewart.

**Data curation:** Brooke H. Nancekivell, Lily W. Martin.

**Formal analysis:** Lily W. Martin, Jill A. Jacobson.

**Funding acquisition:** Brooke H. Nancekivell, Jeremy G. Stewart.

**Investigation:** Brooke H. Nancekivell.

**Methodology:** Brooke H. Nancekivell.

**Project administration:** Jeremy G. Stewart.

**Resources:** Jeremy G. Stewart.

**Software:** Brooke H. Nancekivell.

**Supervision:** Jeremy G. Stewart.

**Visualization:** Lily W. Martin.

**Writing – original draft:** Brooke H. Nancekivell, Lily W. Martin, Jeremy G. Stewart.

**Writing – review & editing:** Brooke H. Nancekivell, Lily W. Martin, Jill A. Jacobson, J. D. Allen, Jeremy G. Stewart.

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
