## [Decision Letter · Decision Letter 0]

25 Jul 2024

PONE-D-24-15386Risky decision-making and nonsuicidal self-injury among young adults: Examining the role of criticism feedbackPLOS ONE

Dear Dr. Stewart,

Thank you for submitting your manuscript to PLOS ONE. After careful consideration, we feel that it has merit but does not fully meet PLOS ONE’s publication criteria as it currently stands. Therefore, we invite you to submit a revised version of the manuscript that addresses the points raised during the review process. Thank you for your patience, I had issues to secure enough reviewers. I have just received the last review today. The reviews are quite positive and the different reviewers' comments should improve the overall manuscript, its clarity and also resolve some current methological issues. Please consider carefully all the reviewers' comments.

We look forward to receiving your revised manuscript.

Kind regards,

Gaëtan Merlhiot

Academic Editor

PLOS ONE

Additional Editor Comments (if provided):

Reviewers' comments:

Reviewer's Responses to Questions

**Comments to the Author**

1. Is the manuscript technically sound, and do the data support the conclusions?

Reviewer #1: Partly

Reviewer #2: Yes

Reviewer #3: Partly

2. Has the statistical analysis been performed appropriately and rigorously? 

Reviewer #1: Yes

Reviewer #2: Yes

Reviewer #3: Yes

3. Have the authors made all data underlying the findings in their manuscript fully available?

Reviewer #1: Yes

Reviewer #2: Yes

Reviewer #3: Yes

4. Is the manuscript presented in an intelligible fashion and written in standard English?

Reviewer #1: Yes

Reviewer #2: Yes

Reviewer #3: Yes

5. Review Comments to the Author

Reviewer #1: The authors, in this paper, examine the association between self-reported frequency of Nonsuicidal Self-Injury (NSSI) and risky decision making on a gambling task. Specifically, they examine the role of negative criticism during the task on the change in disadvantageous decision making for those differing in the frequency of NSSI in the previous month. A number of refinements of methods used in previous studies of this sort were incorporated in the design and the data analysed using a multilevel growth curve model. The authors report three main findings: (a) risky decision making decreased with experience on the task both with and without negative criticism; (b) the decrease was less marked in the negative criticism condition for those reporting two or more instances of NSSI in the month preceding the study; and (c) introduction into the analysis of a number of covariates of NSSI did not alter the inferences to be drawn from the analysis. The authors acknowledge limitations of the study, including with the manipulation of the criticism condition. They conclude that frequency of NSSI is not invariably associated with risky decision making; rather, the association is moderated by the “socioemotional” context, in this case being criticised. The research is, with one exception noted below, technically well done and the report is clearly written. I would also add, although I understand these are not criteria for evaluation, that the report adds importantly to the literature on NSSI and risky decision making and points to potentially useful directions for further work on the topic.

My main concerns with the ms are:

• As the authors themselves identify, their manipulation of the criticism variable introduced a confounding of message content with the occurrence of a message. It may be that receiving a message while completing the gambling task distracts the participant irrespective of its content and this leads to poorer performance by those with higher frequencies of NSSI. This interpretation requires of course that higher frequency NSSI is linked to a greater propensity for distraction. This strikes me as a plausible rival hypothesis to sensitivity to criticism and as such should be at least entertained in the Discussion.

• Crucial to the conclusion reached on the basis of the analysis is that the slopes of the regression lines for the standard gambling task and the gambling task with added criticism differ (p.15, ll. 15-17). How do we know that one is “significantly stronger” that the other? Is it based on a judgement by eye? Is it based on the fact that one is statistically significant and the other is not? Or is there a test of significance of the difference that is not reported. Without the latter, some justification for the inference needs to be provided.

• A total of 23% of the original sample did not provide data for the analysis. I expected some comment on this, but may have missed it.

• Why was frequency of NSSI in the past month selected as the NSSI measure? Would use of any of the other NSSI measures produced the same result?

• The skewness estimates for the variables listed in Table 2 differ substantially. Would this have had any effect on the correlations summarised in the table?

There are, as well, a few of what I think are typological errors that should be addressed:

• Is the final N for analysis 286 or 284. My addition of numbers in Table 1 indicates the latter.

• A correlation of .12 in Table 2 is both statistically significant and non-significant depending on the variables correlated. Does this mean that the Ns differ across variables or that the asterisk has been accidentally omitted for one of the correlations?

• Table 2 provides only one statistic for the SA variable, apart from the correlations. Is there a reason for this? And it is not clear what the statistic that is provided is an estimate of. Is it the average number of suicide attempts across the 286 (284) participants, which seems very high, or is it the total number of attempts for the sample? A median and interquartile range might be a better way of summarising the data for this variable.

Adequate responses to the main concerns listed above would strengthen an already technically sound paper.

Reviewer #2: Thank you for the opportunity to review "Risky decision-making and nonsuicidal self-injury among young adults: Examining the role of criticism feedback" (PONE-D-24-15386). This is a well-written manuscript describing the results of a well-designed experiment investigating a topic of high clinical relevance (i.e., socioemotional contexts impacting risky decision-making among individuals who engage in NSSI). The authors provide evidence to suggest that individuals w/ past-month NSSI histories show particularly poor learning on a risky decision-making task following criticism. The manuscript clearly outlines how the study addresses limitations from previous experiments. Below are some additional comments for the authors to consider that I hope will further enhance this work:

1) P. 3, lines 10-11: This statement seemed a bit of an oversimplification of understandings of NSSI. For example, the Benefits/Barriers Model (Franklin & Hooley, 2018) could be worth mentioning to provide a bit more nuance.

2) P. 8, line 4: Please provide more information on the online advertisements to help better characterize the sample. Where were these advertisements posted? Were there any other inclusion/exclusion criteria applied to those recruited online vs the course credit pool?

3) P. 13, line 20: It would be useful to clarify again that NSSI values referred to behavior frequency over those timeframes (versus dichotomous variables)

4) P. 21, line 23: Please be sure to use person-first language

5) Was there any effort to ascertain how salient participants found the critical statements/how much they identified with the topics? Another explanation for the relatively small effects is that certain participants found the statements harder to identify with and thus experienced them as less threatening.

6) If space allows, it would be helpful to depict the interaction results in a table to make them easier to read.

7) Fig 2 - please revise per APA style to make easier to read

Reviewer #3: The paper observed the possible effect of criticism over decision-making in the population of university students with a high risk for NSSI. The methodology of the research is well-designed and the researchers followed the pre-defined research plan. However, there are some uncertainties concerning basic conceptual issues. In my opinion, they could be resolved by a more detailed description and / or more cautious formulation. I suggest the followings:

1) Title:

As it was highlighted in the limitation section, a very specific population was included (i.e. university students). I suggest to highlight this information even in the title of the article.

2) Participants:

Please provide more information about the recruitment process: e.g. A) how has it been implemented to involve people with higher risk for NSSI, B) what was the target population of online advertisements (ensuring that real persons meeting inclusion criteria have joined to the study).

I also suggest to reconsider the concept of “Table 1”. As it is meant to demonstrate demographical characteristics, participants with no self-injurious behaviour should be highlighted as well. Furthermore, it is not clear how self-injurious behaviours were organized / grouped, since group-numbers are not matched.

3) Measuring decision-making:

As it is mentioned in the “limitations” section, the IGT version used in this study has no audio feedback (Page 23, Line 10), unlike the CGT version. One of my major concerns is that most studies use the IGT with immediate audio and visual feedback (e.g. happy or sad faces and happy or sad sound effects). These effects can be essential - I recommend to see the literature of the “somatic marker hypothesis”. Using an IGT version with no audio effects could be one of the main methodological issues, limiting the comparability of these results with previous findings in the literature. It may also limit the comparability of the IGT and CGT.

4) The concept of measuring the effect of criticism from loved ones:

It is unclear whether the study measured the impact of "criticism from important others" and not the effect of criticism per se. Information about the voice during the CGT was not provided. If the participant chose their father as the reference person, was the voice matched to this choice, or everyone heard the text in the same way?

Although, the mental operation “to imagine” (that the source of the critical comment is a loved one person) may not be a sufficient condition to conclude that indeed the effect of the ‘critical comments from important others’ was measured. Please provide more methodological details if this concept can be supported; otherwise, I suggest to use a more cautious formulation in that question.

5) Modified versions of Iowa Gambling Task:

IGT has more versions (e.g. ABCD and EFGH), enabling to study whether participants are sensitive to reward, punishment or have “myopia for future”. Please take into consideration to discuss decision-making functioning in the light of these research findings.

6) Comparing NSSI and non-NSSI groups:

Direct comparison of persons with history of NSSI and persons with no history of NSSI could strengthen that findings regarding the associations of decision-making performance and criticism may be more specific in case of individuals with history of NSSI.

7) Minor grammatical mistakes:

Some minor grammatical mistakes can be found (punctuation mistakes; unclear sentences, e.g. Page 5, Line 13-14; not precisely used academical English phrases, e.g. “in sum”). Although the text is basically well-written, perhaps a final proofreading would improve the paper's quality.

6. PLOS authors have the option to publish the peer review history of their article (what does this mean?). If published, this will include your full peer review and any attached files.

Reviewer #1: **Yes: **John G. O'Gorman

Reviewer #2: No

Reviewer #3: No

---

## [Author Response · Author response to Decision Letter 0]

20 Aug 2024

We have responded to all reviewer and editor comments in a "Response to Reviewers" document (i.e., "NancekivellEtAl_PLOSONE_ResponseLetter") that was uploaded with other submission documents. We do not duplicate our response here.

---

## [Decision Letter · Decision Letter 1]

1 Oct 2024

Risky decision-making and nonsuicidal self-injury among university students: Examining the role of criticism feedback

PONE-D-24-15386R1

Dear Dr. Stewart,

We’re pleased to inform you that your manuscript has been judged scientifically suitable for publication and will be formally accepted for publication once it meets all outstanding technical requirements.

Kind regards,

Gaëtan Merlhiot

Academic Editor

PLOS ONE

Additional Editor Comments (optional):

Thank you for your thorough revisions and for addressing the reviewers’ comments so thoughtfully. Your responses have clearly strengthened the manuscript, and I am pleased to accept it for publication. I would also like to express my appreciation to the reviewers for their constructive feedback, which has contributed to enhancing the quality of this submission.

Reviewers' comments:

Reviewer's Responses to Questions

**Comments to the Author**

1. If the authors have adequately addressed your comments raised in a previous round of review and you feel that this manuscript is now acceptable for publication, you may indicate that here to bypass the “Comments to the Author” section, enter your conflict of interest statement in the “Confidential to Editor” section, and submit your "Accept" recommendation.

Reviewer #1: All comments have been addressed

Reviewer #3: All comments have been addressed

2. Is the manuscript technically sound, and do the data support the conclusions?

Reviewer #1: Yes

Reviewer #3: Yes

3. Has the statistical analysis been performed appropriately and rigorously? 

Reviewer #1: Yes

Reviewer #3: Yes

4. Have the authors made all data underlying the findings in their manuscript fully available?

Reviewer #1: Yes

Reviewer #3: Yes

5. Is the manuscript presented in an intelligible fashion and written in standard English?

Reviewer #1: Yes

Reviewer #3: Yes

6. Review Comments to the Author

Reviewer #1: (No Response)

Reviewer #3: I thank the Authors for their opennes to the suggestions and thank you for the detailed answers. I have no further suggestions or questions regarding the article.

Thank you for the opportunity to review this paper.

7. PLOS authors have the option to publish the peer review history of their article (what does this mean?). If published, this will include your full peer review and any attached files.

Reviewer #1: No

Reviewer #3: No

---

## [Editor Report · Acceptance letter]

16 Oct 2024

PONE-D-24-15386R1 

PLOS ONE

Dear Dr. Stewart, 

I'm pleased to inform you that your manuscript has been deemed suitable for publication in PLOS ONE. Congratulations! Your manuscript is now being handed over to our production team.

Kind regards, 

on behalf of

Dr. Gaëtan Merlhiot 

Academic Editor

PLOS ONE